# Low-Power pH Sensor Based on Narrow Channel Open-Gated Al_0.25_Ga_0.75_N/GaN HEMT and Package Integrated Polydimethylsiloxane Microchannels

**DOI:** 10.3390/ma13225282

**Published:** 2020-11-22

**Authors:** Xianghong Yang, Jiapei Ao, Sichen Wu, Shenhui Ma, Xin Li, Long Hu, Weihua Liu, Chuanyu Han

**Affiliations:** School of Microelectronics, Xi’an Jiaotong University, Xi’an 710049, China; yxh8000@stu.xjtu.edu.cn (X.Y.); ajp2269228624@stu.xjtu.edu.cn (J.A.); wusichen54@stu.xjtu.edu.cn (S.W.); hasfryma.879@stu.xjtu.edu.cn (S.M.); hulong@xjtu.edu.cn (L.H.); lwhua@mail.xjtu.edu.cn (W.L.); hanchuanyu@mail.xjtu.edu.cn (C.H.)

**Keywords:** Al_0.25_Ga_0.75_N/GaN HEMT, narrow channel, microchannel, pH detection

## Abstract

pH sensors with low-power and strong anti-interference are extremely important for industrial online real-time detection. Herein, a narrow channel pH sensor based on Al_0.25_Ga_0.75_N/GaN high electron mobility transistor (HEMT) with package integrated Polydimethylsiloxane (PDMS) microchannels is proposed. The fabricated device has shown potential advantages in improving stability and reducing power consumption in response to pH changes of the solution. The performance of the pH sensor was demonstrated where the preliminary results showed an ultra-low power (<5.0 μW) at *V*_DS_ = 1.0 V. Meanwhile, the sensitivity was 0.06 μA/V·pH in the range of pH = 2 to pH = 10, and the resolution of the sensor was 0.1 pH. The improvement in performance of the proposed sensor can be related to the narrow channel and microchannel, which can be attributed to better surface Ga_x_O_y_ in a microchannel with larger H^+^ and HO^−^ concentration on the sensing surface during the detection process. The low-power sensor with excellent stability can be widely used in various unattended or harsh environments, and it is more conducive to integration and intelligence, which lays the foundation for online monitoring in vivo.

## 1. Introduction

Biosensor is an interdisciplinary organic combination of bioactive materials (enzymes, proteins, desoxyribonucleic acid (DNA), antibodies, antigens, biofilms, etc.) and electrochemical transducers [1,2,3]. It is an advanced detection method essential for the development of biotechnology and the monitoring method is also fast since the micro-analysis method is at the molecular level of the substance [4]. Biosensor technology will surely be a new growth point between information and biotechnology [5]. It has broad application prospects, namely, clinical diagnosis, industrial control, food and drug analysis (including biopharmaceutical research and development) in the national economy, environmental protection, biotechnology, biochips, and other research areas. With the development of science and technology, biotechnology and electronics infiltrate and merge ulteriorly, therefore more and more electrochemical biosensors have been designed to solve the detection problem of ultra-low concentration [6,7,8].

Among the different kinds of biosensors, aluminum gallium nitride/gallium nitride high electron mobility transistor (AlGaN/GaN HEMT) based devices stand out for their ultra-high sensitivity, fast response speed, and harsh environment adaptability. In addition, it has superior biocompatibility and non-toxicity because of the properties of the III–V nitride materials [9,10,11,12], a controllable surface stoichiometry such as high two-dimensional electronic gas (2DEG) density at the AlGaN/GaN hetero-interface, larger band gap, high breakdown voltage, high electron mobility, and stable chemical properties [13,14,15,16,17]. The AlGaN/GaN HEMT-based biosensor can detect different targets such as ions [18,19,20], DNA [21,22,23], protein [24,25,26], glucose [27,28], prostate specific antigen (PSA) [29,30,31], and cellular response [32], etc. Even without any functionalization, the bare AlGaN surface (open-gated) is especially sensitive to the pH change in the solution. The measurement of pH is needed in many different applications including medicine, biology, chemistry, food science, environmental science, and oceanography. The HEMT-based pH sensor performance depends not only on high-quality AlGaN/GaN heterojunction epitaxial wafers, but also on the structure parameters of the sensitive area and the type and quality of surface sensitive materials. In 2003, G. Steinhof et al. [33] researched the ion sensitivity of native metal oxide on the III-nitride surface, and found a linear response when the pH changed from pH = 2 to pH = 12 for both as-deposited and thermally oxidized GaN surfaces. In 2007, Kang et al. [34] found the use of a Sc_2_O_3_ gate dielectric produced superior results to either a native oxide or UV ozone-induced oxide in the gate region. In 2015, Chen et al. [35] fabricated and studied a GaN-based pH sensor prepared by a hydrogen peroxide (H_2_O_2_) treatment to increase the thickness of the Ga_x_O_y_. In 2015, Lee et al. [27] used a photoelectrochemical (PEC) etching method to grow a ZnO-based nanorod array on the recessed gate of the AIGaN/GaN HEMT sensor to improve device performance. Dong et al. [36] developed an idea to improve the sensitivity of pH sensors based on AlGaN/GaN HEMT by introducing multi-sensing segments in 2018. Li et al. [37] fabricated a normally off AlGaN/GaN HEMT through the photoelectrochemical (PEC) method for a pH sensor application in 2019.

However, the HEMT-based pH sensors reported in the literature have a large channel width-to-length ratio (*W/L*), and does not consider power consumption and environmental interference. Additionally, the larger sensitive area is also not conducive to the miniaturization and integration of sensors. Thus, HEMT-based pH sensor with a particularly small *W*/*L* is highly interesting. The *W*/*L* of the sensitive channel is critical to the performance of the HEMT-based pH sensor. Theoretically, the output current of the HEMT is inversely proportional to the *W* of the device, so a larger sensitive area and larger *W*/*L* can increase the output current and sensitivity. However, long-term exposure of the large sensitive area to the air will reduce the stability and increase interference from environmental uncertainties and power consumption, which are exactly what small channels can avoid, especially the sensitive area of the biosensor that is directly exposed to the liquid under detection. Therefore, it is of great significance if we can not only overcome the difficulty of small channels, but also integrate wide channel devices on the same chip and reduce power consumption and improve anti-interference ability.

Herein, in order to improve the stability and reduce the power consumption of the sensor, we tried to study a narrow channel width open-gated Al_0.25_Ga_0.75_N/GaN HEMT-based pH sensor with package integrated Polydimethylsiloxane (PDMS) microchannels in this paper, which can minimize the external environment interference and ensure the device is not polluted by dust. The unique physical properties and high sensitivity of GaN hetero-junctions to surface charges are used as a tool to distinguish the pH level of the solution. The narrow channel width with microchannels can inherently reduce power consumption and enhance anti-interference. The cost of open-gated is low and changes in sensitive area charge and surface potential are easily controlled by 2DEG and avoid gate leakage power consumption in response to the pH of the solution change. On the other hand, we tried to integrate 30 biosensors on a 2 cm × 2 cm epitaxial substrate. The devices with different channels can be used for pH measurement with different accuracy. The miniaturization of devices is one of the necessary conditions for biochips, and the maturation of the crafts we proposed is an important part of integration, which means to reduce the aspect ratio of the device while ensuring the sensitivity is of great significance. Thus, this research can improve the development of sensors in the direction of miniaturization, integration, and intelligence.

## 2. Basic Structure and Operation Principle

### 2.1. Basic Structure

The schematic configuration of the fabricated open-gated Al_0.25_Ga_0.75_N/GaN HEMT pH sensor is shown in Figure 1. The sensitive area is located between the source and drain, and no metal deposition is required (that is, no Schottky contact is made), so it is called a “open-gated” HEMT pH sensor. The epitaxial structure (NTT Advanced Technology Cor., Tokyo, Japan) is composed of multiple layers of materials by the metal organic chemical vapor deposition (MOCVD) such as silicon substrate, 3.9-μm carbon doped (C-doped) GaN buffer layer, 300-nm GaN layer, 1-nm aluminum nitride (AlN) insert layer, 20-nm Al_0.25_Ga_0.75_N barrier layer, and 2 nm-GaN cap layer. Unlike the traditional HEMT with only AlGaN barrier layer, the GaN channel layer on the substrate, the sensor in this work was fabricated with an AlN insert layer and GaN cap layer. The insert layer can increase the effective conduction band offset of the AlGaN barrier layer and the GaN channel layer. On one hand, it can form a deeper and narrower quantum well because the AlN has a large band gap *E*_g_, which is beneficial to increase the 2DEG concentration in the channel. On the other hand, it can also suppress the disordered scattering of the alloy on the part where 2DEG penetrates into the AlGaN barrier layer, improving the channel electron mobility. The GaN cap layer can increase the Schottky barrier on the AlGaN/GaN heterojunction structure, thus reducing gate leakage and power consumption of the sensor. Ti/Al/Ni/Au (30/180/40/50 nm) was deposited as the source and drain electrode to form an Ohmic contact on the GaN cap layer by electron-beam evaporation. The width and length of the channel sensitive region were *W* (50 μm, 5 μm, and 3 μm) and *L* (900 μm), respectively. This sensitive area was exposed to the tested solution by photolithography and the rest was passivated with SU-8.

### 2.2. Operation Principle

A site-binding model that was initially proposed for the oxide/aqueous sensing mechanism is currently in use to explain the sensing principle for AlGaN/GaN HEMT-based pH sensors [38,39] because a natural oxide layer (such as Al_x_O_y_ and Ga_x_O_y_) is formed on the surface of the AlGaN and GaN layer when the device is exposed in air [33]. Figure 2 presents the operating principle of the proposed HEMT-based pH sensor. According to the site-binding model, the adsorption of protons or hydroxyl ions by surface hydroxyl groups results in positive or negative sites on the Ga_x_O_y_ surface. 2DEG density in the channel is balanced to the surface states of the GaN surface (the cap layer in our case). Therefore, the adsorption of positive or negative charges on the Ga_x_O_y_/GaN surface can change the surface charge state as well as the surface potential, thus altering the 2DEG density [36]. The relevant reactions mechanisms are as follows.
(1)$$\mathrm{MOH}+H^{+}↔\mathrm{MO}H_{2}^{+}$$
(2)$$\mathrm{MOH}+OH^{-}↔MO^{-}$$
where MOH is the hydroxyl groups and M represents Al and Ga due to the Ga-face growth. When testing in the solution of low pH (acid solutions) (i.e., the concentration of H^+^ is larger than that of OH^−^ (*N*_H_^+^ > *N*_OH_^−^)), the GaOH groups tend to accept a proton and become protonated hydroxyls ${\mathrm{GaOH}}_{2}^{+}$ that act as acceptors, represented by Equation (1). Therefore, the GaN surface becomes positively charged [40], leading to the increased sheet carrier density in the 2DEG channel, which finally results in the increase of drain-source current (*I*_DS_). Similarly, when testing in the solution of high pH (alkaline solutions), the concentration of OH^−^ was higher than that of H^+^ (*N*_H_^+^ < *N*_OH_^−^), therefore GaOH groups release proton and become ${\mathrm{GaO}}^{-}$ that acts as donors, represented by Equation (2). Hence, the GaN surface was negatively charged, resulting in the reduced sheet carrier density in the 2DEG channel, which leads to the decreased *I*_DS_. In Figure 2, *I*_DS_ is the initial current of Al_0.25_Ga_0.75_N/GaN HEMT, and $\Delta I_{\mathrm{DS}}$ is the output current of the sensor under a certain pH solution when source-drain voltage (*V*_DS_) is constant. The H^+^ and OH^−^ presented in Figure 2, respectively represent the concentration of hydrogen ions and hydroxide ions in the solution.

## 3. Fabrication Technology

Since the surface of the open-gated HEMT is very close to the Al_0.25_Ga_0.75_N/GaN inter-junction 2DEG channel, the surface charge or potential change in the sensitive area can better control the 2DEG concentration in Al_0.25_Ga_0.75_N/GaN hetero-junction. The output current of the HEMT is inversely proportional to the channel width of the GaN HEMT, and a larger sensitive area and larger *W*/*L* can increase the output current and sensitivity theoretically. However, long-term exposure of the large sensitive area in the air will reduce the stability and increase the interference of water molecules. At the same time, the larger output current will increase the power consumption of the device. Therefore, we designed and manufactured a narrow channel open-gated HEMT-based pH sensor, and the 3D schematic of individual Al_0.25_Ga_0.75_N/GaN HEMT device is shown in Figure 1. Since we used the same sensor layout with previously fabricated nanoribbon-based ion-sensitive field-effect transistors (NR-ISFETs) [41,42], there were also three types of HEMT devices in one chip whose channel lengths were all *L* = 900 μm, and channel widths were *W* = 3 μm, 5 μm, and 50 μm, respectively. Based on an epitaxial wafer on the Si substrate, Ohmic contacts of metals on the GaN cap layer were formed, which are regarded as the source and drain terminals. The main fabrication processes are described in Figure 3, where the detailed manufacturing steps are as follows:
(1)Cleaning the epitaxial wafer: The original Al_0.25_Ga_0.75_N/GaN epitaxial wafer on the Si substrate (NTT Advanced Technology Cor., Figure 3a) is composed of a p-type low resistivity Si substrate, GaN buffer layer (3.9 μm-C-doped), GaN layer (300 nm), AlN layer (1 nm), Al_0.25_Ga_0.75_N barrier layer (20 nm), and GaN cap layer (2 nm). Ultrasonically clean with acetone, isopropanol, and ethanol for 3 min, rinse with a large amount of deionized water, and blow dry with nitrogen.(2)Ion implantation to achieve isolation: First use AZ-5214E photoresist lithography to prepare ion implantation patterns, where the detailed steps are: (1) photoresist spin coating: keep the homogenizer at 1000 rpm for 10 s and 2000 rpm for 30 s, respectively; (2) pre-baking: keep the device at 110 °C for 70 s; (3) align exposure: exposure time of 7 s; (4) Developing: Immerse the device in in a solution of AZ-400K: deionized water = 1:3 for 60 s; (5) ion implantation: The ion implantation process was completed at the Institute of Semiconductors, Chinese Academy of Sciences. The ion type was Ar^+^, the energy was 30/40/60/80 keV in turn, and the dose was 5.0 × 10^13^/cm^2^ (Figure 3b).(3)Ohmic contact and pads: After cleaning the wafer, another photolithography was conducted (which lithography process is the same as that of ion implantation lithography), followed by the deposition of Ti/Al/Ni/Au (30/180/40/50 nm) in sequence by electron-beam evaporation (Ohmiker-50BR, Cello Technology Co. Ltd., Taiwan, China). The pads were prepared by magnetron sputtering (MSP-300B, Weinaworld Co. Ltd., Beijing, China), Ti/Au: 20/100 nm. Then, the lift-off process was carried out to form the Ohmic metal contacts of the source/drain regions, interconnect, and pads (Figure 3c).(4)Finally, a 2μm thick SU-8 photoresist layer (GM 1040, Gersteltec Sarl. Inc., Pully, Switzerland) was coated and patterned as the top passivation layer to prevent the metals from erosion by the liquid during testing (Figure 3d), and the sensing regions over the channel and only the pads were exposed. The process is listed as follows: (1) photoresist spin coating: keep the homogenizer at 1600 rpm for 40 s; (2) relax the device for 5 min at room temperature; (3) pre-baking: keep the device at 65 °C for 5 min, then 95 °C for 5 min, then slowly cool down to 23 °C, cooling time should be longer than 2 h; (4) exposure: exposure time is 32 s; (5) post-baking: keep the device at 65 °C for 5 min, then 95 °C for 5 min, then slowly cool down to 23 °C, cooling time should be longer than 4 h; (6) developing: immerse the device in propylene glycol monomethyl ether acetate (PGMEA) developer for 1 min; (7) hard-baking: keep the device at 135 °C for 2 h.

The photograph of the open-gated Al_0.25_Ga_0.75_N/GaN HEMT-based pH sensor after fabrication is shown in Figure 4a. Figure 4b presents the microscope image of the sensor chip, from the top to the bottom of the figure, where the channel widths were *W* = 50 μm, 5 μm, and 3 μm.

## 4. Results and Discussion

The pH sensitive characteristics of the proposed sensor were studied by the revolving microprobe stage and semiconductor parameter analyzer (Keithley 4200A-SCS), as shown in Figure 5a. Figure 5b shows the fabricated open-gated Al_0.25_Ga_0.75_N/GaN HEMT chip integrated with PDMS-based microchannels for pH sensing, this includes outlet and inlet parts for acidic, alkaline, and neutral solutions and a polymethyl methacrylate (PMMA) removable plate for fixing and replacing the chip. The PDMS microchannels are shown in Figure 5c, which were fabricated by coating and patterning of SU-8 on a Si substrate with a thickness of 10 µm.

The injection and withdrawal of the pH solution uses a medical syringe with controllable flow. After the open-gated Al_0.25_Ga_0.75_N/GaN HEMT-based pH sensor system was built, we performed the input and output characteristics (*I*_DS-_*V*_DS_ without pH solutions) of Al_0.25_Ga_0.75_N/GaN HEMT with different channel widths (*W* = 3 μm, 5 μm, 50 μm) under channel length *L* = 900 μm first, which is shown in Figure 6. In the range of *V*_DS_ = 0 to 2.0 V, the *I*_DS_ increased as *W*/*L* increased, and the sensor had a larger output current when *W* = 50 μm compared with the other two sensors, which is consistent with the theory.

Although a larger *W* means that the sensitive channel area of the sensor is increased, it will also increase the instability of the sensor due to the exposed air, so that it cannot meet the detection of polar solutions in various unattended or harsh environments. The output signal of the sensor is small and can be amplified by the subsequent signal processing circuit, but its stability and anti-interference ability are derived from its own performance.

Subsequently, we studied the pH sensitivity of the HEMT-based sensor. The different pH solutions were obtained by diluting deionized water with hydrochloric acid (HCl) and sodium hydroxide (NaOH), which were used in the liquid testing and delivered to open-gated HEMT sensitive surface through the microchannels. According to theory and experiment, the output current of the Al_0.25_Ga_0.75_N/GaN HEMT is inversely proportional to the *W*, a larger *W*/*L*, and larger sensitive area can increase the output current and sensitivity, but long-term exposure of the large sensitive area to the air will reduce the stability due to increased interference from environmental uncertainties and power consumption. Therefore, only the testing results of the device at *W* = 3 μm are demonstrated here as those results are representative. Figure 7a shows the current as a function of bias voltage from HEMTs with Ga_2_O_3_ in the gate region exposed to a series of solutions whose pH varied from 2 to 10 when the temperature was 20 ± 2 °C and humidity <85%.

The adsorption of polar molecules on the surface of the HEMT affected the surface potential and device characteristics, and the current was significantly increased upon exposure to these polar liquids as pH decreased. The *I*_DS_ in each case can be extracted and plotted as a function of pH value (Figure 7b), and the sensitivity was 0.06 μA/V·pH and the maximum accuracy was 0.1 pH. Figure 7 shows that the HEMT-based pH sensor with the Ga_2_O_3_ gate dielectric is sensitive to the concentration of the polar liquid and therefore could be used to differentiate between liquids into which a small amount of leakage of another substance has occurred.

Figure 8 is the static characteristics of the open-gated Al_0.25_Ga_0.75_N/GaN HEMT-based pH sensor under different *V*_DS_ and different pH, which are the linearity, hysteresis, and repeatability characteristics of the sensor for three cycles of forward (input-increased) and backward (input-decreased) calculated using MATLAB. The experimental results showed that under a constant pH = 7, *V*_DS_ changed from 0 V to 2.0 V and 2.0 V to 0 V, the sensor had excellent nonlinearity, repeatability, and hysteresis characteristics, which were 0.06%F.S., 0.14%F.S., and 0.56%F.S., respectively (Figure 8a). This indicates that the sensor could work under different bias voltages and had excellent stability. Furthermore, when the pH values changed from 2 to 10 and 10 to 2, the nonlinearity, hysteresis, and repeatability were 15.33%F.S., 4.28%F.S., and 8.92%F.S. at *V*_DS_ = 1.0 V (Figure 8b), respectively. The result was not as perfect as the result in Figure 8a, which may be affected by the defects and surface states of the natural metal oxide Ga_2_O_3_ material on the sensitive surface of the sensor [43,44], but this does not affect the sensor’s suitability for repeated measurements with different pH values and satisfying different changes in acid and alkaline solutions.

The real-time *I*_DS_ measurements were also performed with the changing pH solutions, as shown in Figure 9. The *I*_DS_ of the sensor in acid solution (pH = 4), neutral solution (pH = 7), and alkaline solution (pH = 10) was measured at *V*_DS_ = 1.0 V, respectively. Each pH value is measured every 5 s for a total of 50 measurements. From the figure, the sensor shows excellent sensitivity and stability characteristics, which is attributed to the narrow channel of the Al_0.25_Ga_0.75_N/GaN HEMT sensor and the integrated packaging microchannel to avoid the larger sensitive area from being disturbed by environmental interference and to ensure the quality of the oxide in the sensitive area and larger H^+^ and HO^−^ concentration. Furthermore, according to the test statistics, after more than 200 measurements, the characteristics of the pH sensor could be restored to 99.36% of the initial current after being cleaned with DI water and placed at room temperature for 1 h.

In order to further promote the development of monolithic integrated multi-function sensors and integrated circuits (IC), the development of a low power consumption sensor device is highly important in the future. Therefore, the research into low-power sensors becomes more and more important with the development of IC manufacturing technology. Reducing the output current is a means to achieve low power consumption, so the sensor in this article used a smaller sensitive area width to length ratio. At the same time, a GaN cap layer was added to the basic structure of the sensor, which could increase the mobility of 2DEG at the expense of a slight decrease in carrier concentration under the polarization effect.

On the other hand, the GaN cap layer could increase the Schottky barrier on the AlGaN/GaN heterojunction structure, thereby significantly reducing gate leakage [45]. Of course, the sensor designed in this article was an open-gated structure, so there was no gate leakage and gate power consumption. Figure 10 shows the power consumption of the Al_0.25_Ga_0.75_N/GaN HEMT-based pH sensor under different *V*_DS_ in the range of pH = 2 to 10. When *V*_DS_ = 0.1 V, 1.0 V, and 2.0 V, the average value of *P* was 0.18 μW, 3.62 μW, and 14.5 μW, respectively. The small size and low power consumption of this sensor enabled the small-sized substrate to easily accommodate multiple sensors to monitor the pH, temperature, humidity, and air quality of the environment.

Table 1 gives the main characteristic parameters of the Al_0.25_Ga_0.75_N/GaN HEMT-based pH sensor and mainly includes the key geometric structure parameters (W, integration) and performance parameters (Sensitivity-*S*_pH_, Nonlinearity-*N*_L_, Hysteresis-*H*, Repeatability-*R*, Power consumption-*P*, Detection range-*D*_r_, Resolution-*r*, and lifetime-*L*_t_).

## 5. Conclusions

Al_0.25_Ga_0.__75_N/GaN HEMT-based pH sensors with open gate and narrow channel width were fabricated and characterized. The open-gated HEMT could improve the control ability of the hetero-junction channel 2DEG by the change in charge and potential in the sensitive area. Additionally, the narrow channel sensor has a relatively small output current, which can reduce its power consumption. The pH sensitivity of the sensor could reach 0.06 μA/V·pH in the range of pH = 2 to 10, resolution was 0.1 pH, and it had ultra-low power (<5.0 μW) and small hysteresis in multiple measurements at *V*_DS_ = 1.0 V. Moreover, the performance of the HEMT-based pH sensor system could be improved in a microchannel, which could be attributed to better surface Ga_x_O_y_ in a microchannel with larger H^+^ and HO^-^ concentration on the sensing surface. The sensitivity of sensors with the narrow channel was slightly inferior than that of the sensors with a wide channel. However, this kind of sensor with a narrow channel has the virtue of lower power consumption and excellent stability, which can be widely used in various unattended and harsh environments. Moreover, the features of integration and intelligence provide unlimited prospects for in-body online monitoring.

## Figures and Tables

**Figure 1 materials-13-05282-f001:**
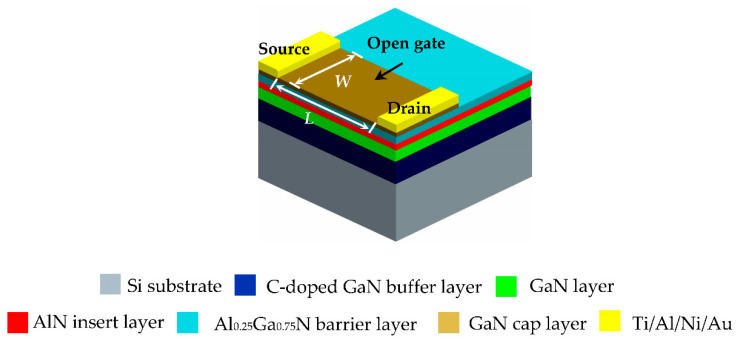
The schematic configuration of the open-gated Al_0.25_Ga_0.75_N/GaN HEMT-based pH sensor.

**Figure 2 materials-13-05282-f002:**
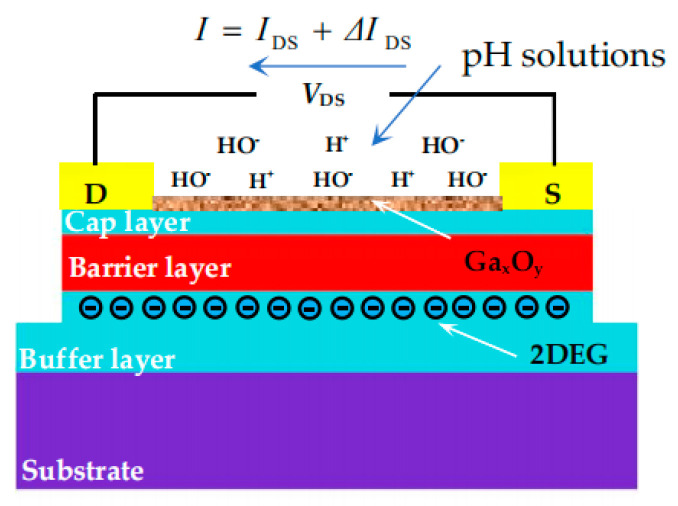
The operation principle of the open-gated Al_0.25_Ga_0.75_N/GaN HEMT-based pH sensor.

**Figure 3 materials-13-05282-f003:**
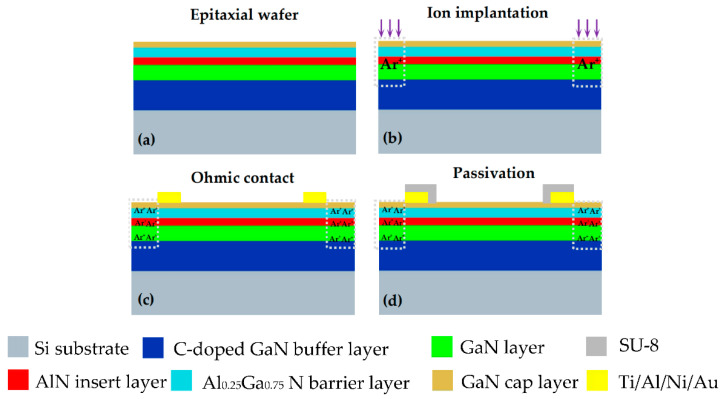
The fabrication process of the open-gated Al_0.25_Ga_0.75_N/GaN HEMT-based pH sensor: (**a**) GaN epitaxial wafer on the Si substrate; (**b**) Ion implantation; (**c**) Ohmic contact metal: Ti/Al/Ni/Au; (**d**) SU-8 passivation layer.

**Figure 4 materials-13-05282-f004:**
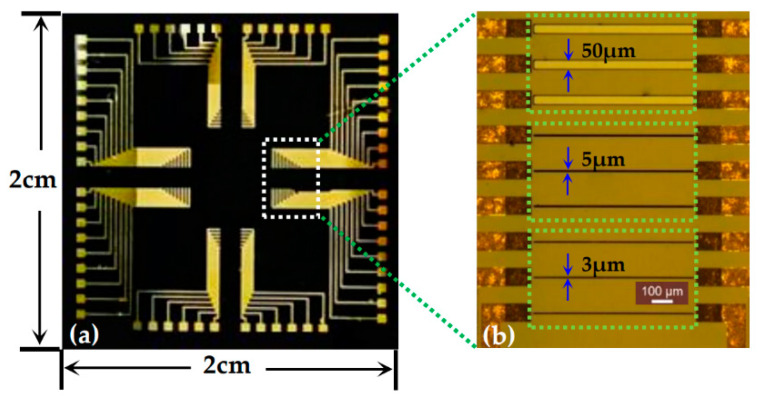
The photograph of the open-gated Al_0.25_Ga_0.75_N/GaN HEMT-based pH sensor: (**a**) photograph of the sensor chip; (**b**) microscope image of the sensor chip.

**Figure 5 materials-13-05282-f005:**
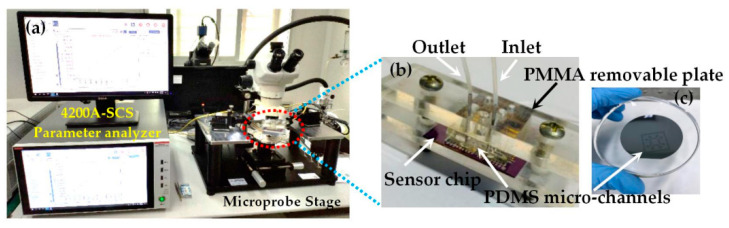
The testing platform of the open-gated Al_0.25_Ga_0.75_N/GaN HEMT-based pH sensor: (**a**) testing platform; (**b**) photograph of the packaged pH sensor; (**c**) polydimethylsiloxane (PDMS)-based microchannels.

**Figure 6 materials-13-05282-f006:**
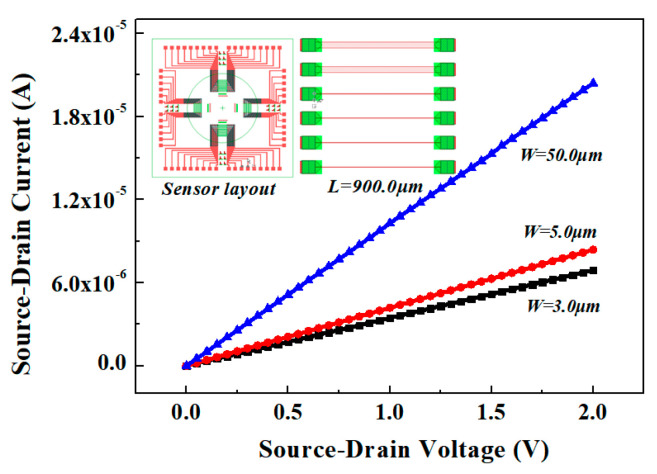
The output characteristics (*I*_DS_-*V*_DS_) of Al_0.25_Ga_0.75_N/GaN HEMT with different channel widths (*W* = 50 μm, 5 μm, 3 μm) without pH solutions.

**Figure 7 materials-13-05282-f007:**
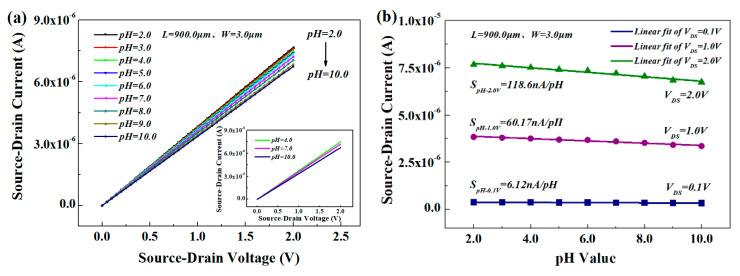
The sensitive characteristics of open-gated Al_0.25_Ga_0.75_N/GaN HEMT-based sensor (*W* = 3 μm) in different pH solutions: (**a**) *I*_DS_-*V*_DS_ with different pH values; (**b**) *I*_DS_-pH with different *V*_DS_ values.

**Figure 8 materials-13-05282-f008:**
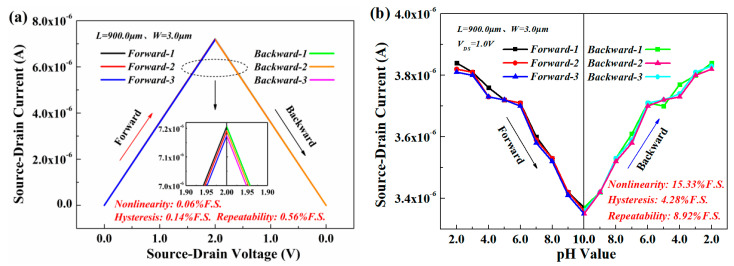
The analysis of the static characteristics of the open-gated Al_0.25_Ga_0.75_N/GaN HEMT-based pH sensor: (**a**) *I*_DS_-*V*_DS_ at pH = 7.0; (**b**) *I*_DS_-pH at *V*_DS_ = 1.0 V.

**Figure 9 materials-13-05282-f009:**
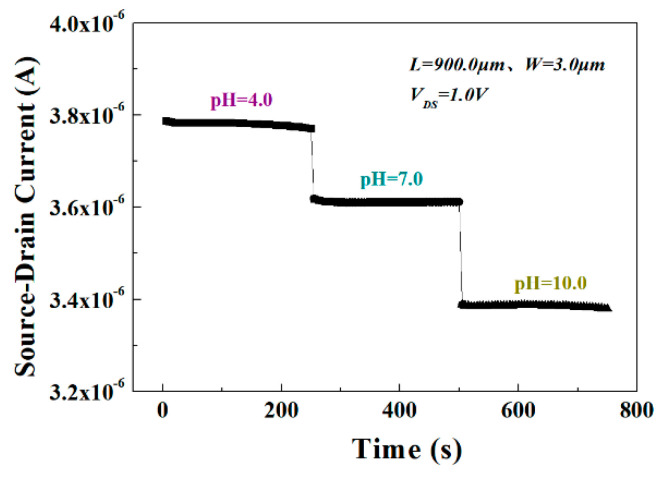
*I*_DS_ response of the open-gated Al_0.25_Ga_0.75_N/GaN HEMT-based pH sensor during real-time measurement of the pH value from 4 to 10 at *V*_DS_ = 1.0 V with the time interval of 5 s.

**Figure 10 materials-13-05282-f010:**
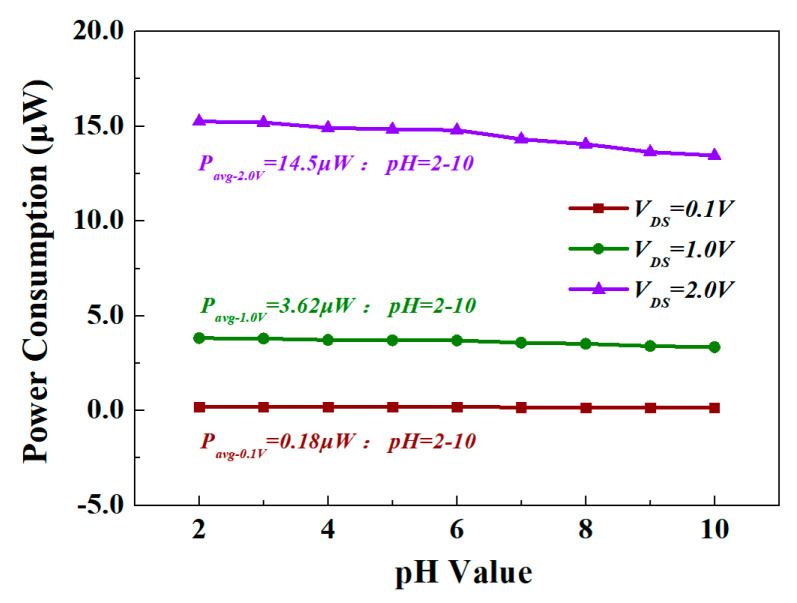
The power consumption of the A_0.25_Ga_0.75_N/GaN HEMT-based pH sensor under different *V*_DS_ in the range of pH = 2 to 10.

**Table 1 materials-13-05282-t001:** The main characteristic parameters of the Al_0.25_Ga_0.75_N/GaN HEMT-based pH sensor.

Main Characteristics Parameter (*V*_DS_ = 1.0V)		Values	Units
Channel width	*W*	3	μm
Integration		30	
Sensitivity	S_pH_	0.06	μA/V·pH
Nonlinearity	*N* _L_	15.33	%F.S.
Hysteresis	*H*	4.28	%F.S.
Repeatability	*R*	8.92	%F.S.
Power consumption	*P*	<5.0	μW
Detection range	*D* _r_	2.0–10.0	pH
Resolution	*r*	0.1	pH
Lifetime	*L_t_*	>200	times

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
