# Peer review of "Low-Power pH Sensor Based on Narrow Channel Open-Gated Al0.25Ga0.75N/GaN HEMT and Package Integrated Polydimethylsiloxane Microchannels"

_materials, 2020, doi:10.3390/ma13225282_

Round 1

Reviewer 1 Report

This Manuscript deals with  The improvement in performance of pH sensor based on Narrow Channel Open-Gated Al0.25Ga,0.75N/GaN HEMT and Package Integrated PDMS.This MS was so interesting however to get it publishable, the authors needs to adress the following concerns:

Please make sure that all sentences that are not the authors’ work have appropriate references: (e.g., line 25-31 page 1 need some references).

Abbreviations need to be used scientifically, please make sure that this journal guidelines have been followed properly. e.g., AlGaN/GaN HEMT seems to be introduced first in full.

Lines 58-84 were not clear for me. I couldn’t clearly understand why the authors decided to improve the proposed sensor through reduceing the power consumption and at the same time, have an improved stability. please make sure that there is a reasonable relation between sentences and paragraphs.

the last sentence of the the abstract section says something that seems different from the last sentence of conclusion section. Please reconsider those to have no conflict.

‘’Basic structure’’ section should have more details about the equipments, applied materials, and methodologies used in order to achieve that layered structure.

Like wise, the ‘’ Fabrication technology’’ also needed some more in detail about the fabrication. It was not clear to me, the relation between the sentences was missing in some where’s, e.g., ‘’line 139-153.

the mechanical properties of this recommended sensor have not evaluated.

how the statistical significance of their results have been evaluated.

Author Response

Dear editor,

Following the comments of the reviewer, the manuscript of "Low-Power pH Sensor Based on Narrow Channel Open-Gated Al0.25Ga0.75N/GaN HEMT and Package Integrated Polydimethylsiloxane Micro-channels" has been carefully revised. Main modifications are listed as follow:

Comments and Suggestions for Authors (Reviewer 1):

  1. Please make sure that all sentences that are not the authors’ work have appropriate references: (e.g., line 25-31 page 1 need some references).

Response:

Thank you very much for your kindly remind, we have added references in the first paragraph of the introduction. Please see lines 25-31 for details.

  • Huber A, Demartis S, Neri D. The use of biosensor technology for the engineering of antibodies and enzymes. J Mol Recognit, 1999, 12, 198-216.
  • Toldra A, Alcaraz C, Diogene J, et al. Detection of Ostreopsis cf. ovata in environmental samples using an electrochemical DNA-based biosensor. Sci Total Environ, 2019, 689, 655-661.
  • Atci E, Babauta JT, Sultana ST, et al. Microbiosensor for the detection of acetate in electrode-respiring biofilms. Biosens Bioelectron, 2016, 81, 517-523.
  • Maduraiveeran G, Sasidharan M, Ganesan V. Electrochemical sensor and biosensor platforms based on advanced nanomaterials for biological and biomedical applications. Biosens Bioelectron 2018, 103, 113-129.
  • Vigneshvar S, Sudhakumari CC, Senthilkumaran B, et al. Recent Advances in Biosensor Technology for Potential Applications - An Overview. Front Bioeng Biotech, 2016, 4.
  • Alonso-Lomillo MA, Dominguez-Renedo O. Screen-Printed Biosensors in Drug Analysis. Curr Pharm Anal, 2017, 13, 169-174.
  • Marrakchi M, Dzyadevych SV, Namour P, et al. A novel proteinase K biosensor based on interdigitated conductometric electrodes for proteins determination in rivers and sewers water. Sensor Actuat B-Chem, 2005, 111, 390-395.
  • Adaszewska A, Kalinska-Bienias A, Jagielski P, et al. The use of BIOCHIP mosaics in diagnostics of bullous pemphigoid: Evaluation and comparison to conventional multistep procedures. J Cutan Pathol, 2020, 47, 121-127.
  1. Abbreviations need to be used scientifically, please make sure that this journal guidelines have been followed properly. e.g., AlGaN/GaN HEMT seems to be introduced first in full.

Response:

Thank you very much for your comments, full names for all abbreviations at first time mentioned have been provided in paper. Details are as follows:

  • Desoxyribonucleic acid (DNA)
  • Aluminum Gallium Nitride/Gallium Nitride high electron mobility transistor (AlGaN/GaN HEMT)
  • Two-dimensional electronic gas (2DEG)
  • Prostate specific antigen (PSA)
  • Hydrogen peroxide (H2O2)
  • Photoelectrochemical (PEC)
  • Metal organic chemical vapor deposition (MOCVD)
  • Carbon doped (C-doped)
  • Aluminium Nitride (AlN)
  • Nanoribbon-based ion-sensitive field-effect transistors (NR-ISFETs)
  • Propylene glycol monomethyl ether acetate (PGMEA)
  • Polymethyl methacrylate (PMMA)
  • Hydrochloric acid (HCl)
  • Sodium hydroxide (NaOH)
  • Integrated circuits (IC)
  1. Lines 58-84 were not clear for me. I couldn’t clearly understand why the authors decided to improve the proposed sensor through reducing the power consumption and at the same time, have an improved stability. please make sure that there is a reasonable relation between sentences and paragraphs.

Response:

We are very sorry that the original lines 58-84 of the article is not clear and thank you very much for your comments and Suggestions. We have completed the modification and see lines 62-89 for details. The modification is as follows:

However, the HEMT-based pH sensors reported in the literature have a large channel width-to-length ratio (W/L), and does not consider power consumption and environmental interference. Additionally, the larger sensitive area is also not conducive to the miniaturization and integration of sensors.Thus, HEMT-based pH sensor with a particularly small W/L is highly interesting. The W/L of the sensitive channel is critical to the performance of the HEMT-based pH sensor. Theoretically, the output current of the HEMT is inversely proportional to the W of the device, a larger sensitive area and larger W/L can increase the output current and sensitivity. But long-term exposure of the large sensitive area to the air will reduce the stability, increase interference from environmental uncertainties and power consumption, which are exactly what small channels can avoid, especially the sensitive area of the biosensor that is directly exposed to the liquid under detection. Therefore, it is of great significance if we can not only overcome the difficulty of small channel, but also integrate wide channel devices on the same chip and reduce power consumption and improve anti-interference ability.

Herein, in order to improve the stability and reduce the power consumption of the sensor, we tried to study a narrow channel width open-gated Al0.25Ga0.75N/GaN HEMT-based pH sensor with package integrated PDMS micro-channels in this paper, which can minimize the external environment interference and ensure the device is not polluted by dust. The unique physical properties and high sensitivity of GaN hetero-junctions to surface charges are used as a tool to distinguish the pH level of solution. The narrow channel width with micro-channels can inherently reduce power consumption and enhance anti-interference. The cost of open-gated is low and changes of sensitive area charge and surface potential are easily controlled by 2DEG and avoid gate leakage power consumption in response to the pH of the solution change. On the other hand, we try to integrate 30 biosensors on a 2cm×2cm epitaxial substrate. The devices with different channels can be used for pH measurement with different accuracy. The miniaturization of devices is one of the necessary conditions for biochips, and the maturation of the crafts we proposed is an important part of integration, which mean to reduce the aspect ratio of the device while ensuring the sensitivity is of great significance. Thus, this research can improve the development of sensors in the direction of miniaturization, integration and intelligence.

  1. The last sentence of the abstract section says something that seems different from the last sentence of conclusion section. Please reconsider those to have no conflict.

Response:

We are very sorry that the last sentence of the abstract section says something that seems different from the last sentence of conclusion section of the article is not clear and thank you very much for your comments and Suggestions. We have completed the modification and see lines 306-318 for details. The modification is as follows:

Al0.25Ga0.75N/GaN HEMT-based pH sensor with open gate and narrow channel width have been fabricated and characterized. The open-gated HEMT can improve the control ability of the hetero-junction channel 2DEG by the change of charge and potential in the sensitive area. And the narrow channel sensor has a relatively small output current, which can reduce its power consumption. The pH sensitivity of the sensor can reach 0.06 μA/V·pH in the range of pH=2 to 10, resolution is 0.1 pH, and it has ultra-low power (<5.0 μW) and small hysteresis in multiple measurements at VDS=1.0 V. Moreover, the performance of the HEMT-based pH sensor system can be improved in a micro-channel, which may be attributed to better surface GaxOy in a microchannel with larger H+ and HO- concentration on the sensing surface. The sensitivity of sensors with narrow channel is slightly inferior than that of sensors with wide channel. However, this kind of sensor with narrow channel has the virtue of lower power consumption and excellent stability, which can be widely used in various unattended and harsh environments. Moreover, the features of integration and intelligence provide unlimited prospects for in-body online monitoring.

  1. “Basic structure’’ section should have more details about the equipment, applied materials, and methodologies used in order to achieve that layered structure.

Response:

Thanks for your comments. The preparation of GaN requires extremely high process conditions, despite it has excellent properties as the third-generation semiconductor. Because GaN grown in the laboratory has high defect density, which cannot meet the requirements of good performance for sensors. Therefore, the GaN epitaxial wafer used in this work was purchased from NTT Advanced Technology Cor. (Japan). We have completed the modification according to your comments and suggestions, please refer to line 92-110 for details:

The schematic configuration of the fabricated open-gated Al0.25Ga0.75N/GaN HEMT pH sensor is shown in Figure.1. The sensitive area is located between the source and drain, and no metal deposition is required (that is, no Schottky contact is made), so it’s called a "open-gated" HEMT pH sensor. The epitaxial structure (NTT Advanced Technology Cor., Japan) is composed of multiple layers of materials by metal organic chemical vapor deposition (MOCVD) such as Silicon substrate, 3.9-μm Carbon doped (C-doped) GaN buffer layer, 300-nm GaN layer, 1-nm Aluminium Nitride (AlN) insert layer, 20-nm Al0.25Ga0.75N barrier layer and 2nm-GaN cap layer. Different from the traditional HEMT with only AlGaN barrier layer, GaN channel layer on the substrate, the sensor in this work is fabricated with AlN insert layer and GaN cap layer. The insert layer can increase the effective conduction band offset of the AlGaN barrier layer and the GaN channel layer. On the one hand, it can form a deeper and narrower quantum well because of the AlN has a large band gap Eg, which is beneficial to increase the 2DEG concentration in the channel. On the other hand, it can also suppress the disordered scattering of the alloy on the part where 2DEG penetrates into the AlGaN alloy barrier layer, improving the channel electron mobility. The GaN cap layer can increase the Schottky barrier on the AlGaN/GaN heterojunction structure, thus reducing gate leakage and power consumption of the sensor. Ti/Al/Ni/Au (30/180/40/50nm) is deposited as source and drain electrode to form an ohmic contact on the GaN cap layer by electron-beam evaporation. The width and length of channel sensitive region are W (50μm, 5μm and 3μm) and L (900μm), respectively. This sensitive area was exposed to the tested solution by photolithography and the rest was passivated with SU-8.

  1. Like wise, the ‘’Fabrication technology’’ also needed some more in detail about the fabrication. It was not clear to me, the relation between the sentences was missing in some where’s, e.g., ‘’line 139-153’’. 

Response:

Thanks for your comments. The detailed preparation process of the sensor has been supplemented in the “Fabrication technology’’, see line 162-192 for details:

The mainly fabrication processes are described in Figure 3, detailed manufacturing steps are as follows:

(1) Cleaning the epitaxial wafer: The original Al0.25Ga0.75N/GaN epitaxial wafer on Si substrate (NTT Advanced Technology Cor., Figure 3.(a)) is composed of p-type low resistivity Si substrate, GaN buffer layer (3.9 μm-C-doped), GaN layer (300 nm), AlN layer (1 nm), Al0.25Ga0.75N barrier layer (20 nm) and GaN cap layer (2 nm). Ultrasonically clean with acetone, isopropanol, and ethanol for 3 minutes, rinse with a large amount of deionized water and blow dry with nitrogen.

(2) Ion implantation to achieve isolation: first use AZ-5214E photoresist lithography to prepare ion implantation patterns, the detailed steps are: 1) photoresist spin coating: the homogenizer are kept at 1000 rpm for 10 seconds and 2000 rpm for 30 seconds, respectively; 2) pre-baking: keep the device at 110℃ for 70 seconds; 3) align exposure: exposure time is 7 seconds; 4) developing: Immerse the device in in a solution of AZ-400K: deionized water = 1:3 for 60 seconds; 5) ion implantation: The ion implantation process was completed in the institute of semiconductors, Chinese academy of sciences. The ion type is Ar+, the energy is 30/40/60/80 keV in turn, and the dose is 5.0×1013/cm2 (Figure. 3(b)).

(3) Ohmic contact and pads: after cleaning the wafer, another photolithography was conducted (which lithography process is the same as that of ion implantation lithography), followed by the deposition of Ti/Al/Ni/Au (30/180/40/50 nm) in sequence by electron-beam evaporation (Ohmiker-50BR, Cello Technology Co., Ltd.). And the pads is prepared by magnetron sputtering (MSP-300B, Weinawolrd Co., Ltd.), Ti/Au: 20/100 nm. Then the lift-off process was carried out to form the Ohmic metal contacts of source/drain regions, interconnect and pads (Figure. 3(c)). 

(4) Finally, a 2μm thick SU-8 photoresist layer (GM 1040, Gersteltec Sarl. Inc.) was coated and patterned as the top passivation layer to prevent the metals from erosion by the liquid during testing (Figure.3 (d)), and the sensing regions over the channel and the pads are exposed only. The process is listed as follows: 1) photoresist spin coating: the homogenizer is kept at 1600 rpm for 40 seconds; 2) relaxing the device for 5 minutes at room temperature; 3) pre-baking: keep the device at 65℃ for 5 minutes, then 95℃ for 5 minutes, then slowly cooling down to 23℃, cooling time should be larger than 2h; 4) exposure: exposure time is 32 seconds; 5) post-baking: keep the device at 65℃ for 5 minutes, then 95℃ for 5 minutes, then slowly cooling down to 23℃, cooling time should be larger than 4h; 6) developing: immerse the device in propylene glycol monomethyl ether acetate (PGMEA) developer for 1 minute; 7) hard-baking: keep the device at 135℃ for 2 hours.

The photograph of the open-gated Al0.25Ga0.75N/GaN HEMT-based pH sensor after fabricated as shown in Figure 4(a). The Figure 4(b) is the microscope image of the sensor chip, from the top to the bottom of the figure, the channel width are W=50 μm, 5 μm, 3 μm.

  1. The mechanical properties of this recommended sensor have not evaluated.

Response:

Thanks for your attention to the mechanical performance of GaN sensor of this paper. We assume the mechanical performance of this sensor may come from two parts: one is the mechanical properties of the flexible sensor based on GaN on the flexible substrate; the other is the effect of the stress applied to the GaN sensor on the 2DEG. For starters, we did not pay attention to the mechanical performance of flexible GaN sensor in this work. Because it is extremely difficult to grow GaN directly on a flexible substrate, small-area GaN flexible devices can be realized unless transfer technology was adopted. In order to obtain high consistency of the pH sensors, we choose a Si-based substrate instead of a flexible substrate in this work, and fabricated multiple sensors on one substrate. On the other hand, the 2DEG at the AlGaN/GaN heterojunction interface is mainly due to the spontaneous polarization and piezoelectric polarization effects of GaN and AlGaN. When the material is subjected to tensile and compressive stresses, the 2DEG concentration will changes, resulting in a change in the output current of HEMT device. Based on this effect, AlGaN/GaN heterojunction can be applied to pressure sensors in combination with flexible substrates (See the references below). However, no relevant reports have been reported in the application of biosensors. Therefore, flexible substrate based GaN biosensor is our future research direction.

  • Tan X , Lv Y J , Zhou X Y , et al. AlGaN/GaN pressure sensor with a Wheatstone bridge structure[J]. AIP Advances, 2018, 8(8):085202.
  • Durga G , Ifat J , Goutam K . High Temperature AlGaN/GaN Membrane Based Pressure Sensors[J]. Micromachines, 2018, 9(5):207.
  • Zhu J , Zhou X , Jing L , et al. Piezotronic Effect Modulated Flexible AlGaN/GaN High-Electron-Mobility Transistors[J]. ACS Nano, 2019.
  • Yang L , Duan B , Dong Z , et al. The analysis model of AlGaN/GaN HEMTs with electric field modulation effect[J]. IETE Technical Review, 2019:1-12.
  • Wang A , Zeng L , Wang W , et al. Static and dynamic simulation studies on the AlGaN/GaN pressure sensor[J]. Semiconductor ence and Technology, 2019, 34(11).
  • Tan X , Lv Y , Zhou X , et al. High performance AlGaN/GaN pressure sensor with a Wheatstone bridge circuit[J]. Microelectronic Engineering, 2020, 219(Jan.):111143.1-111143.4.
  1. How the statistical significance of their results have been evaluated.

Response:

Thank you very much for your comments dear reviewer. 30 pH sensors with channel width as 50μm, 5μm and 3μm (10 sensors for each width) were fabricated on 2×2cm2 Si-based GaN epitaxial wafer. After preliminary tested, 5 sensors have no electrical properties (we believe this should be attribute to the 2DEG channel blocking when the sensors were isolated by ion implantation), and the electrode metal was damaged during the late packaging for other 3 sensors. Therefore, the yield rate of our pH sensor is around 70%. Among the remaining 22 devices, we selected the best devices for 50μm, 5μm and 3μm to study their I-V characteristics (see Fig. 6).

Reviewer 2 Report

The manuscript „ Low-Power pH Sensor Based on Narrow Channel  Open-Gated Al0.25Ga0.75N/GaN HEMT and Package Integrated Polydimethylsiloxane micro-channels " by Xin Li et al. s an interesting study on the preparation and performance characteristics of the microsensor. The designs of this type of sensors have been known for many years and the materials used in them and their characteristics do not constitute a sufficient scientific novelty worth publishing in "Materials". Moreover, the subject of the manuscript is beyond the scope of this journal and would not attract the attention of readers who are rather looking for information about new materials and their properties. A description of the sensor design, based on a microchannel system with its detailed characteristics, would be more appropriate, for example, to "Sensors" and I would suggest to redirect this manuscript there. The text is well written,  an introductory part includes a review of most of the latest works, but omits some important publications on the subject of AlGaN / GaN open-gate HEMT devices. The results are presented carefully, the figures are legible and correctly described, but their discussion should be deepened on the basis of literature data from research on similar systems (e.g. in Sensors 2011, 11, 3067-3077, and others). The authors suggest that the sensor they constructed could be used to test biological samples, but there are no test results confirming such assumptions, and there is no information about the possible impact of viscosity and composition of biological fluids on the accuracy of determinations. The conclusions are presented too briefly, they should be expanded to highlight  the most important elements of the novelty of the described solution.

Author Response

Dear editor,

Following the comments of the reviewer, the manuscript of "Low-Power pH Sensor Based on Narrow Channel Open-Gated Al0.25Ga0.75N/GaN HEMT and Package Integrated Polydimethylsiloxane Micro-channels" has been carefully revised. Main modifications are listed as follow:

Comments and Suggestions for Authors (Reviewer 2)

The manuscript "Low-Power pH Sensor Based on Narrow Channel Open-Gated Al0.25Ga0.75N/GaN HEMT and Package Integrated Polydimethylsiloxane Micro-channels " by Xin Li et al. s an interesting study on the preparation and performance characteristics of the microsensor. The designs of this type of sensors have been known for many years and the materials used in them and their characteristics do not constitute a sufficient scientific novelty worth publishing in "Materials". Moreover, the subject of the manuscript is beyond the scope of this journal and would not attract the attention of readers who are rather looking for information about new materials and their properties. A description of the sensor design, based on a microchannel system with its detailed characteristics, would be more appropriate, for example, to "Sensors" and I would suggest to redirect this manuscript there.

Response:

Dear reviewer, thank you very much for your recognition of this article and the recommendation of the sensors journal. Although our goal is to design and implement biosensors, a complete biosensor also includes subsequent interface circuits, including information transmission, processing, display, and control.So, on second thought, we decided that this work was more suitable for the "Materials" journal, because we mainly proposed the material and structure of the sensor.

In this paper, the characteristics of the sensor are improved by adding insert layer material AlN and cap layer material GaN on the basis of traditional AlGaN/GaN HEMT, the insert layer AlN matertials can increase the effective conduction band offset of the AlGaN barrier layer and the GaN channel layer. On the one hand, it can form a deeper and narrower quantum well because of the AlN has a large band gap Eg, which is beneficial to increase the 2DEG concentration in the channel. On the other hand, it can also suppress the disordered scattering of the alloy on the part where 2DEG penetrates into the AlGaN barrier layer, improving the channel electron mobility. The GaN cap layer materials can increase the Schottky barrier on the AlGaN/GaN heterojunction structure, thus reducing gate leakage and power consumption of the sensor.. The pH solution was detected by using PDMS to prepare micro-channels. The detailed protocols for fabrication of PDMS-based microchannels are listed below. 1) Mixture preparation: Mix the silicone elastomer base and curing agent (SYLGARD 184, World Precision Instruments) by mass ratio of 10:1 and stir thoroughly; 2) Degassing treatment I: Degas the PDMS mixture in the drying oven for 2.5 h in room temperature; 3) Silanization treatment: Pipetting a few drops of trimethylchlorosilane (TMCS) onto the surface of prepared SU-8 molding substrate; 4) Molding: Put the SU-8 substrate in a petri dish and pour the PDMS mixture slowly into the petri dish (about 3 mm thick); 5) Degassing treatment II: Put the petri dish in the drying oven for 1.5 h in room temperature for degassing; 6) Incubation: Incubate the PDMS mixture with SU-8 molding in oven at 50 ℃ for 12 h; 7) Stripping: Peel off the PDMS layer from the petri dish after it is curing. The photographs of fabricated PDMS layer before and after stripping from the patterned SU-8 substrate are shown in Figure. 5-c.

 The text is well written, an introductory part includes a review of most of the latest works, but omits some important publications on the subject of AlGaN/GaN open-gated HEMT devices.

Response:

Dear reviewer, thank you very much for your suggestions and comments.

AlGaN/ Gan-based open-gated HEMT is mainly used in biosensor. The sensitive area of this structure is located above the channel, and changes in surface charge or surface potential can easily affect the AlGaN/GaN interface concentration of 2DEG, which means that the gate control capability is very strong. In this paper, references are added to the open-gated HEMT-based sensors.

 The results are presented carefully, the figures are legible and correctly described, but their discussion should be deepened on the basis of literature data from research on similar systems (e.g. in Sensors 2011, 11, 3067-3077, and others).

Response:

Dear reviewer, thank you very much for your suggestions and comments.According to research reports, pH sensors based on GaN HEMT mainly focus on (1) the preparation of pH-sensitive materials, such as single-medium-sensitive membranes and composite-medium-sensitive membranes (2) the design of sensitive membrane structures and device structures (3) pH identification methods (4) research on sensitization mechanism and other aspects. Although natural oxide is used as the pH-sensitive film in this paper, a GaN cap layer and an AlN insertion layer are added to the sensor structure design. The insert layer can increase the effective conduction band offset of the AlGaN barrier layer and the GaN channel layer. On the one hand, it can form a deeper and narrower quantum well because of the AlN has a large band gap Eg, which is beneficial to increase the 2DEG concentration in the channel. On the other hand, it can also suppress the disordered scattering of the alloy on the part where 2DEG penetrates into the AlGaN barrier layer, improving the channel electron mobility. The GaN cap layer can increase the Schottky barrier on the AlGaN/GaN heterojunction structure, thus reducing gate leakage and power consumption of the sensor. In addition, we reduced the W/L of the device and packaged integrated PDMS micro-channels to further reduce the power consumption of the sensor and improve the anti-interference ability, this is different from many literature reports (e.g. in Sensors 2011, 11, 3067-3077). At the same time, some articles reported that the influence of light on the sensitivity of the sensor was not considered in the pH detection process, but In this paper, it is tested entirely in a dark environment. Actually, the AlGaN/GaN HEMT has been known to generate photocurrent from 650 nm (1.91 eV) to 325 nm (3.81 eV) of the UV-visible spectrum. Considering the bandgap of GaN of 3.4 eV, electron-hole pairs are generated as carriers when irradiating light having energy higher than energy bandgap. However, when irradiating light that does not have energy above bandgap (red, green, blue light), carrier concentration also increases. In this case, the increasing rate is smaller than another case when electron-hole pair was newly generated by UV-light. Therefore, this can be explained by electron detrapping in trap inside the device. Electron trapped inside device was activated by light and join to carrier, increasing the current. Various studies could investigate the de-trapping mechanism which was calculated by Arrhenius’ equation for electron activation energy, electron recapturing energy [S-Ref.1-3].

  • 1] Woo K , Kang W , Lee K , et al. Enhancement of cortisol measurement sensitivity by laser illumination for AlGaN/GaN transistor biosensor[J]. Biosensors & Bioelectronics, 2020, 159:112186.

[S-Ref.2] Cheney, D. (Invited) AlGaN/GaN HEMT Reliability and Trap Detection Using Optical Pumping[J]. ECS Transactions, 2014, 61(4):197-204.

[S-Ref.3] Ahn, Shihyun, et al. "Effect of proton irradiation dose on InAlN/GaN metal-oxide semiconductor high electron mobility transistors with Al2O3 gate oxide." Journal of Vacuum Science & Technology B, Nanotechnology and Microelectronics: Materials, Processing, Measurement, and Phenomena 34.5 (2016): 051202.

 The authors suggest that the sensor they constructed could be used to test biological samples, but there are no test results confirming such assumptions, and there is no information about the possible impact of viscosity and composition of biological fluids on the accuracy of determinations.

Response:

Dear reviewer, we are very sorry that we did not use this sensor to detect other biological analytes. Because this paper focuses on the design and verification of new sensor materials and new sensor structures. Therefore, we just adopt the most basic pH index as the verification basis.The influence of the reviewer on the "viscosity and composition of biofluids" is very important.We will carry out research in further work.

 The conclusions are presented too briefly, they should be expanded to highlight  the most important elements of the novelty of the described solution.

Response:

Thank you very much for your kindly remind, we have completed the modification and see lines 306-318 for details. The modification is as follows:

Al0.25Ga0.75N/GaN HEMT-based pH sensor with open gate and narrow channel width have been fabricated and characterized. The open-gated HEMT can improve the control ability of the hetero-junction channel 2DEG by the change of charge and potential in the sensitive area. And the narrow channel sensor has a relatively small output current, which can reduce its power consumption. The pH sensitivity of the sensor can reach 0.06 μA/V·pH in the range of pH=2 to 10, resolution is 0.1 pH, and it has ultra-low power (<5.0 μW) and small hysteresis in multiple measurements at VDS=1.0 V. Moreover, the performance of the HEMT-based pH sensor system can be improved in a micro-channel, which may be attributed to better surface GaxOy in a microchannel with larger H+ and HO- concentration on the sensing surface. The sensitivity of sensors with narrow channel is slightly inferior than that of sensors with wide channel. However, this kind of sensor with narrow channel has the virtue of lower power consumption and excellent stability, which can be widely used in various unattended and harsh environments. Moreover, the features of integration and intelligence provide unlimited prospects for in-body online monitoring.

Reviewer 3 Report

This reviewer recommends the solution of some 3-4-letters abbreviations.

Also the exact fabrication methods are missing, how the nm-wide layers has been formed.

The ion implantation experimental description should have also been included among others.

On Fig 4. the 50-5-3 um-s should point to some places of the image or should have different labelling.

Does the authors have information on the lifetime of a sensor? How many times can it be used? Does the sensitivity give information on the precisity? How many decimals can be determined e.g. can it make difference between pH 7.65 and 7.70?

Author Response

Dear editor,

Following the comments of the reviewer, the manuscript of "Low-Power pH Sensor Based on Narrow Channel Open-Gated Al0.25Ga0.75N/GaN HEMT and Package Integrated Polydimethylsiloxane Micro-channels" has been carefully revised. Main modifications are listed as follow:

Comments and Suggestions for Authors (Reviewer 3)

  1. This reviewer recommends the solution of some 3-4-letters abbreviations. 

Response:

Thank you very much for your comments, full names for all abbreviations at first time mentioned have been provided. Details are as follows:

  • Desoxyribonucleic acid (DNA)
  • Aluminum Gallium Nitride/Gallium Nitride high electron mobility transistor (AlGaN/GaN HEMT)
  • Two-dimensional electronic gas (2DEG)
  • Prostate specific antigen (PSA)
  • Hydrogen peroxide (H2O2)
  • Photoelectrochemical (PEC)
  • Metal organic chemical vapor deposition (MOCVD)
  • Carbon doped (C-doped)
  • Aluminium Nitride (AlN)
  • Nanoribbon-based ion-sensitive field-effect transistors (NR-ISFETs)
  • Propylene glycol monomethyl ether acetate (PGMEA)
  • Polymethyl methacrylate (PMMA)
  • Hydrochloric acid (HCl)
  • Sodium hydroxide (NaOH)
  • Integrated circuits (IC)
  1. Also the exact fabrication methods are missing, how the nm-wide layers has been formed. 

Response:

Thank you very much for your comments dear reviewer. We have completed the modification according to your comments and suggestions, please refer to line 183-192 for details:

The nano channel in this work is realized by photolithography, which mainly includes photoresist spin coating, pre-baking, exposure, post-baking, developing and hard baking. In this work, SU-8 photoresist was used to form the passivation protection to fabricate nano channel, the primary process is listed as follows: 1) Photoresist spin coating: the homogenizer is kept at 1600rpm for 40 seconds; 2) Relaxing the device for 5 minutes; 3) Pre-baking: keep the device at 65℃ for 5 minutes, then 95℃ for 5 minutes, then slowly cooling down to 23℃, cooling time should be larger than 2h; 4) Exposure: exposure time is 32 seconds; 5) Post-baking: keep the device at 65℃ for 5 minutes, then 95℃ for 5 minutes, then slowly cooling down to 23℃, cooling time should be larger than 4h; 6) Developing: Immerse the device in propylene glycol monomethyl ether acetate (PGMEA) developer for 1 minute; 7) Hard-baking: keep the device at 135℃ for 2 hours. A narrow channel (sensitive region) device can be formed by the above process steps.

  1. The ion implantation experimental description should have also been included among others. 

Response:

We have completed the modification according to your comments and suggestions, please refer to line 169-176 for details.

Ion implantation to achieve isolation: first use AZ-5214E photoresist lithography to prepare ion implantation patterns, the detailed steps are: 1)Photoresist spin coating: the homogenizer are kept at 1000rpm for 10 seconds and 2000 rpm for 30 seconds, respectivelyï¼›2)Pre-baking: keep the device at 110℃ for 70 secondsï¼› 3)Align exposure: exposure time is 7 seconds; 4)Developing: Immerse the device in in a solution of AZ-400K:DI water = 1:3 for 60 seconds. 5)Ion implantation: The ion implantation process was completed in the Institute of semiconductors, Chinese Academy of Sciences, the injected ion type was Ar+, the energy was 30/40/60/80 keV in turn, and the injected dose was 5.0×1013/cm2 (Figure. 3(b)).

  1. On Fig 4. the 50-5-3 um-s should point to some places of the image or should have different labelling.

Response:

Following the comments, we have modified Figure 4.

Figure 4. The photograph of the open-gated Al0.25Ga0.75N/GaN HEMT-based pH sensor: (a) photograph of the sensor chip; (b) microscope image of the sensor chip.

  1. Does the authors have information on the lifetime of a sensor? How many times can it be used? Does the sensitivity give information on the precisity? How many decimals can be determined e.g. can it make difference between pH 7.65 and 7.70?

Response:

Thanks very much for your comments. Long lifetime and high precision are crucial parameters for sensors in practical applications. According to the test statistics, after more than 200 times measurements, the characteristics of the pH sensor can be restored to 99.36% of the initial characteristics after being cleaned with DI water and placed at room temperature for 8 hours. In addition, the maximum accuracy of the sensor is 0.1pH, and the measurement of 0.05pH cannot be achieve so far. Detailed results illustrated in Figure S-1. We will also focus on improving the sensitivity and precision of the sensor by optimizing sensitive materials in the future.

Figure S-1. Resolution curve of the open-gated Al0.25Ga0.75N/GaN HEMT-based pH sensor during real-time measurement.
